# Analysis of Both Lipid Metabolism and Endocannabinoid Signaling Reveals a New Role for Hypothalamic Astrocytes in Maternal Caloric Restriction-Induced Perinatal Programming

**DOI:** 10.3390/ijms22126292

**Published:** 2021-06-11

**Authors:** Rubén Tovar, Antonio Vargas, Jesús Aranda, Lourdes Sánchez-Salido, Laura González-González, Julie A. Chowen, Fernando Rodríguez de Fonseca, Juan Suárez, Patricia Rivera

**Affiliations:** 1Instituto de Investigación Biomédica de Málaga-IBIMA, 29010 Málaga, Spain; rubentovar7@hotmail.com (R.T.); antoniovargasfuentes@gmail.com (A.V.); jesu95aranda@gmail.com (J.A.); lourdesmedia25@hotmail.com (L.S.-S.); lauragonzalez_13@hotmail.com (L.G.-G.); fernando.rodriguez@ibima.eu (F.R.d.F.); 2UGC Salud Mental, Hospital Regional Universitario de Málaga, 29010 Málaga, Spain; 3Andalucia Tech, Facultad de Medicina, Universidad de Málaga, Campus de Teatinos s/n, 29071 Málaga, Spain; 4Department of Endocrinology, Instituto de Investigación Biomédica la Princesa, Fundación Investigación Biomédica del Hospital Infantil Universitario Niño Jesús, 28009 Madrid, Spain; julieann.chowen@salud.madrid.org; 5CIBEROBN (Centro de Investigación Biomédica en Red Sobre Fisiopatología de la Obesidad y Nutrición), Instituto de Salud Carlos III, 28009 Madrid, Spain; 6IMDEA Food Institute, CEI UAM + CSIC, 28009 Madrid, Spain; 7Departamento de Anatomía Humana, Medicina Legal e Historia de la Ciencia, Universidad de Málaga, 29071 Málaga, Spain

**Keywords:** perinatal programming, hypothalamus, astrocytes, endocannabinoid system, lipid metabolism

## Abstract

Maternal malnutrition in critical periods of development increases the risk of developing short- and long-term diseases in the offspring. The alterations induced by this nutritional programming in the hypothalamus of the offspring are of special relevance due to its role in energy homeostasis, especially in the endocannabinoid system (ECS), which is involved in metabolic functions. Since astrocytes are essential for neuronal energy efficiency and are implicated in brain endocannabinoid signaling, here we have used a rat model to investigate whether a moderate caloric restriction (R) spanning from two weeks prior to the start of gestation to its end induced changes in offspring hypothalamic (a) ECS, (b) lipid metabolism (LM) and/or (c) hypothalamic astrocytes. Monitorization was performed by analyzing both the gene and protein expression of proteins involved in LM and ECS signaling. Offspring born from caloric-restricted mothers presented hypothalamic alterations in both the main enzymes involved in LM and endocannabinoids synthesis/degradation. Furthermore, most of these changes were similar to those observed in hypothalamic offspring astrocytes in culture. In conclusion, a maternal low caloric intake altered LM and ECS in both the hypothalamus and its astrocytes, pointing to these glial cells as responsible for a large part of the alterations seen in the total hypothalamus and suggesting a high degree of involvement of astrocytes in nutritional programming.

## 1. Introduction

In recent years, a large number of studies in humans and animals have supported the idea that exposure to certain injuries during critical periods of development can increase the risk of developing diseases in later life such as respiratory diseases, cancer, psychiatric pathologies, obesity and the metabolic syndrome. These injuries can be varied, and include drugs, stress and diet, with the latter being the most studied in metabolic and behavioral programming, a phenomenon called nutritional programming [1,2,3]. The most studied period has been the prenatal period. However, the programming could be more extensive, including the pre-conceptional period as well as lactation and even childhood and puberty, in which the individual may be especially vulnerable [4,5,6]. Initial programming studies have focused on caloric restriction during pregnancy, showing subsequent alterations in offspring related to metabolic syndrome such as increased adiposity, impaired glucose metabolism and dyslipidemia [7].

The alterations induced by nutritional programming in the hypothalamus of offspring are of special relevance given the important role of this brain region in energy homeostasis, appetite and body composition. Maternal malnutrition during the pre/gestational period can affect many hypothalamic signaling systems controlling appetite and energy expenditure [8,9,10]. One of them is the endocannabinoid system, which is involved in numerous functions both in development and adult life, highlighting energy metabolism, feeding behavior, emotional control and neurodegenerative diseases [11,12,13,14,15].

Diet can modify the endocannabinoid system (ECS) in different tissues, including the brain during development [16,17,18]. Maternal caloric restriction implemented during the preconceptional/pregnancy period decreases endocannabinoid and/or NAE levels in the hypothalamus and hippocampus in male offspring at birth, being associated with overweight, increased adiposity and anxiety-related responses later in the adulthood [12,17,19]. Furthermore, it has been shown that changes in nutritional programming resulting from maternal diet restriction during fetal development can modify the expression of ECS elements later in life and lead to long-lasting impacts on energy metabolism (lipid and cholesterol metabolism) in a sex-dependent manner [14].

The two main endocannabinoids, anandamide (AEA) and 2-arachidonoylglycerol (2-AG), are derived from arachidonic acid (AA), which in turn is a derivative of linoleic acid, an essential *n*-6 fatty acid in the diet [20]. Therefore, maternal malnutrition can alter endocannabinoid signaling and lipid metabolism (LM), inducing metabolic and neurobehavioral alterations throughout the life of the offspring [11,14,21].

It is well known that brain metabolic support is provided by astrocytes, structural intermediary cells that act as conduits between blood vessels and neurons. Thus, astrocytes provide blood borne nutrients such as glucose, monocarboxylates and fatty acids to neurons. In addition, the metabolic plasticity of astrocytes is essential for neuronal energy efficiency [22,23].

Recent data have indicated that astrocytes are critical regulators of nervous system development [24]. Thus, astrocytes participate in neurodevelopment both at a structural and functional level, producing signals that regulate synapses and inflammatory processes. In relation to this, it has been shown that early alterations that involve inflammatory processes, such as certain perinatal injuries (malnutrition, stress, inflammation), involve the participation of astrocytes and their relationship with certain neurological and/or neurodegenerative diseases in adulthood [25,26,27].

Recently, Abbink and colleagues [28] reviewed initial evidence that astrocytes are affected by early life adversity (ALS) in rodents, undergoing morphological and metabolic changes that could affect neural function. However, the vast majority of studies have focused on structural rather than functional changes in astrocytes, with a focus on the expression of GFAP without elucidating its functional consequences. Furthermore, a functional alteration caused by ALS is not always linked to a marked change in the astrocyte marker GFAP [28].

Recent evidence has shown that astrocytes are involved in endocannabinoid signaling, responding to exogenous cannabinoids and endocannabinoids through the activation of cannabinoid type 1 (CB1) receptors, which release intracellular calcium and stimulate glutamate synaptic transmission and plasticity [29]. In addition, astrocytes actively participate in cannabinoid signaling, generating their own endocannabinoids as well as the fatty acids that give rise to them [21,23,30]. In a previous study, we showed that perinatal malnutrition induced by a hypercaloric diet produces hypothalamic alternations in adult offspring, both in endocannabinoid signaling and in lipid metabolism of this brain region. We also demonstrated the role of hypothalamic astrocytes in these changes through their response to palmitic acid and anandamide in culture [21].

Elucidation of the contributions of astrocytes in neurodevelopment will benefit our mechanistic understanding of perinatal programming by providing new preventive and/or therapeutic pathways for neurological and metabolic conditions throughout life. Therefore, in the present study we analyzed the effects of a moderate caloric restriction (20%) implemented during both a pregestational period and throughout the subsequent gestation on LM and cannabinoid signaling in the hypothalamus of PND2 pups. We also carried out primary cultures of hypothalamic astrocytes from the offspring to analyze the gene and protein expression of the same components that were previously analyzed in the hypothalamus.

Our results reveal an important participation of hypothalamic astrocytes in perinatal programming, pointing to them as promising targets for neuroregulation and/or neuroprotection. We show that most of the alterations in LM and endocannabinoid signaling in the hypothalamus of PND2 offspring induced by maternal caloric restriction are also found in astrocytes, confirming their role in perinatal metabolic programming.

## 2. Results

### 2.1. Effects of Caloric Restriction on Maternal Body Weight

Cumulative body weights in rat dams from the diet restriction group were significantly lower than controls before pregnancy (pregestation period) and during gestation. These differences are particularly evident from the second week of caloric restriction onward (*/** *p* < 0.05/0.01; Figure 1).

During the overall gestational period, calorie-restricted dams also gained body weight, but it was delayed when compared with control rats (*** *p* < 0.001; Figure 1).

### 2.2. Effects of Maternal Caloric Restriction on Markers of Gliosis and Neuroinflammation in PND2 Offspring Hypothalamus

A diet effect was found on the mRNA levels of *Gfap* [F (1,32) = 4.804; *p* < 0.05] and *vimentin* [F (1,29) = 9.950; *p* < 0.01], with an overall increase in *Gfap* and *vimentin* mRNA in the hypothalamus of animals (males and females) born to caloric-restricted mothers (# *p* < 0.05; Figure 2A,B).

There was an effect of sex [F (1,32) = 6.308; *p* < 0.05] and an interaction [F (1,32) = 6.881; *p* < 0.05] between diet and sex on the mRNA levels of *Il6*, with an increase in *Il6 mRNA* in the hypothalamus of female offspring born to caloric-restricted mothers compared to restricted males and control females (*/# *p* < 0.05; Figure 2D).

An interaction between diet and sex on the mRNA levels of *Il4* was also found [F (1,32) = 7.705; *p* < 0.01] (Figure 2F).

We also found an effect of sex [F (1,32) = 9.727; *p* < 0.01] and an interaction [F (1,32) = 6.702; *p* < 0.05] between diet and sex on the mRNA levels of *Il10*, with an increase in *Il10 mRNA* in the hypothalamus of female offspring born to caloric-restricted mothers compared to males (** *p* < 0.01; Figure 2G).

Two-way ANOVA indicated an interaction between maternal diet and sex [F (1,20) = 5.024; *p* < 0.05] on the protein levels of GFAP (Figure 2J).

There was a diet effect on vimentin protein levels [F (1,20) = 4.927; *p* < 0.05] (Figure 2K).

No effects were found on the mRNA and protein levels of Iba1, a marker of microglia (Figure 2C,L).

### 2.3. Effects of Maternal Caloric Restriction on Lipid Metabolism and Endocannabinoid Signaling in PND2 Offspring Hypothalamus

Regarding lipid metabolism, there was a diet effect on *Scd1* [F (1,32) = 16.26; *p* < 0.01] and *Acox1* [F (1,32) = 20.20; *p* < 0.001] mRNA levels, with an increase in *Scd1* and *Acox1* in male and female offspring born to mothers with caloric restriction (#/##/### *p* < 0.05/0.01/0.001; Figure 3B,D).

We also found a diet effect on the mRNA levels of the fatty acid synthesizing enzyme *Fasn* [F (1,32) = 4.667; *p* < 0.05] and regulators of lipid homeostasis *Srbf1* and *Srebf2* [F (1,32) > 9.689; *p* < 0.01], with the maternal caloric restriction increasing *Srebf1* and *Srebf2* mRNA levels (#/## *p* < 0.05/0.01; Figure 3F,G).

Regarding the endocannabinoid system, there was a maternal diet effect on the mRNA levels of the synthesizing enzymes *Dagla* and *Daglb* [F (1,32) = 8.414; *p* < 0.01] and degrading enzyme *Magl* [F (1,32) = 5.983; *p* < 0.05] of 2-AG, with an increase in *Dagla* and *Daglb* in the hypothalamus of female offspring born to mothers with caloric restriction compared to females born to mothers with a control diet (# *p* < 0.05; Figure 3J,K).

A maternal diet effect was also found on the mRNA levels of the AEA synthesizing enzyme *Nape-pld* [F (1,32) = 7.588; *p* < 0.01] (Figure 3M).

Regarding protein expression, we only found an effect of the maternal diet on the expression of DAGLb [F (1,20) = 7.045; *p* < 0.05], with the maternal caloric restriction decreasing DAGLb protein levels in the hypothalamus of male offspring compared to control males (# *p* < 0.05; Figure 4K,O).

### 2.4. Effects of Maternal Caloric Restriction on Lipid Metabolism and Endocannabinoid Signaling in Hypothalamic Astrocytes

There was an effect of diet [F (1,39) = 5.962; *p* < 0.05] and an interaction [F (1,39) = 5.046; *p* < 0.05] between diet and sex on the mRNA levels of *Cpt1a*, with an increase in *Cpt1a* in astrocytes from female offspring born to caloric-restricted mothers compared to control female astrocytes (# *p* < 0.05; Figure 5A).

The mRNA levels of *Acox1* and *Fasn* were also affected by maternal diet [F (1,39) > 4.223; *p* < 0.05], with maternal caloric restriction increasing *Fasn* in male astrocytes compared to control male astrocytes (# *p* < 0.05; Figure 5E).

A sex effect was found on *Srebf1* mRNA expression [F (1,39) = 4.133; *p* < 0.05]. Astrocytes from female offspring born to control mothers had a higher level of *Srebf1* than control male astrocytes (* *p* < 0.05; Figure 5F).

Regarding the endocannabinoid system, maternal caloric restriction had an effect on the mRNA levels of *Dagla* [F (1,39) = 9.243; *p* < 0.01], with astrocytes from female offspring born to caloric-restricted mothers showing a higher level of *Dagla* than control female astrocytes (## *p* < 0.01; Figure 5J).

An interaction between maternal diet and sex was found on *Magl* [F (1,39) = 26.75; *p* < 0.001], astrocytes from female offspring born to caloric-restricted mothers had higher levels of *Magl* compared to control male astrocytes. However, maternal caloric restriction increased *Magl* in female astrocytes compared to control female astrocytes (## *p* < 0.01; Figure 5L).

The mRNA levels of *Nape-pld* and *Faah* were also affected by maternal diet [F (1,39) > 4.461; *p* < 0.05]. A sex effect [F (1,39) = 33.87; *p* < 0.001] and an interaction between factors [F (1,39) = 17.06; *p* < 0.001] were also found on *Faah* mRNA expression. Astrocytes from female offspring born to caloric-restricted mothers had a higher level of *Faah* than control female astrocytes (### *p* < 0.001; Figure 5N). We also observed a decrease in *Faah* mRNA expression in control female astrocytes compared to female offspring born to control mothers (*** *p* < 0.001; Figure 5N).

Regarding the endocannabinoid system, a sex effect was found on the protein levels of CB1 and DAGLa [F (1,20) > 5.346; *p* < 0.05]; astrocytes from female offspring born to caloric-restricted mothers had a higher level of DAGLa than restriction male astrocytes (* *p* < 0.05; Figure 6I). The protein levels of DAGLb were also affected by sex and maternal diet [F (1,20) > 4.342; *p* < 0.05] (Figure 6J).

An interaction between maternal diet and sex was found for MAGL [F (1,20) = 4.858; *p* < 0.05] (Figure 6K).

## 3. Discussion

The present study demonstrates that a moderate caloric restriction during the maternal period (pregestation and gestation) induces changes in the expression of both mRNA and proteins of several factors involved in lipid metabolism (LM) and cannabinoid signaling in the hypothalamus of male and female offspring at birth (PND2). The effect of the maternal diet increasing the overall mRNA expression of *Gfap* and *Vimentin,* but not *Iba-1,* in response to caloric restriction, suggests that hypothalamic astrocytes, but not microglial cells, could be involved in these changes. Based on these findings, we studied the expression of lipid/endocannabinoid signaling in hypothalamic astrocytes of pups born to restricted-caloric mothers, with the aim of elucidating the role of astrocytes in developmental programming. In general, our results show that a moderate caloric restriction during pregestation and gestation has an effect on the LM of the hypothalamic astrocytes of the offspring, altering the gene and/or protein expression of enzymes involved in fatty acid oxidation (Cpt1a and Acox1), as well as that of the lipogenic enzyme Fasn. In addition, astrocytic cannabinoid signaling in offspring is also altered by low maternal calorie intake, as revealed by the food restriction-dependent expression of the enzymes for both the synthesis (DAGLa, DAGLb and NAPE-PLD) and degradation (MAGL and FAAH) of the main endocannabinoids. Interestingly, many of these changes could explain the alterations induced by maternal caloric restriction in LM and cannabinoid signaling of the total hypothalamus of the offspring. These results point to astrocytes as fundamental elements in perinatal metabolic programming.

The caloric restriction implemented during pregnancy is related to metabolic alterations of the offspring in both the short and long term. These alterations include increased adiposity, alterations in glucose metabolism, lipid signaling and dyslipidemia, among others [3,7,31]. The association of maternal caloric restriction with enhanced expression of endocannabinoid synthesis enzyme (DAGL and NAPE-PLD) points to an overactivity of the ECS that might be responsible for ulterior changes in metabolic phenotype [12]. It is interesting to highlight that adult overactivity of the ECS also results in increased adiposity, alterations in glucose metabolism and dyslipidemia [32,33]

Given that the hypothalamus is the regulatory center for metabolism, it is conceivable that the metabolic alterations induced in offspring by maternal malnutrition are partly due to abnormal hypothalamic development in the offspring [9,34]. Indeed, it has been shown that both high and low caloric intake during the perinatal stage induces changes in the number of orexigenic/anorexigenic neurons, affects the hypothalamic signaling of leptin and insulin, and DNA methylation of hypothalamic genes that are implicated in the future risk of metabolic disease in the offspring [9,34,35,36,37,38].

The endocannabinoid system (ECS) is a lipid signaling system that consists of two receptors (CB1 and CB2), endogenous or endocannabinoid ligands (mainly anandamide and 2-AG) and endocannabinoid synthesis and degradation enzymes. The ECS regulates many aspects of neurodevelopment and energy homeostasis, controlling energy balance and LM centrally in the hypothalamus and peripherally (adipocytes, liver, muscle), interacting with numerous anorexigenic and orexigenic pathways [39].

The ECS is present in the brain from the first stages of development and is expressed in a specific pattern throughout the different stages, which indicates a regulatory role in neurodevelopment. In the early stages of development, ECS-mediated actions involve axonal growth, fasciculation and the establishment of correct neuronal connectivity [40]. Disruption of the optimal functioning of the ECS in critical periods can lead to mental health problems later in life. For example, neurodevelopmental stress induced by maternal deprivation is associated with dysregulation of the ECS that may be relevant to schizophrenia and other psychiatric neurodevelopmental disorders [41].

Given that the perinatal period is a time of maximum sensitivity to sex hormones, the induced alterations in neuronal circuits during neurodevelopment are often sex-dependent. In this sense, the effects of this neonatal stress on the endocannabinoid system were more evident in males than in females, and since these sexual dimorphisms were found at an early neonatal age they could be connected to the organizational effects of gonadal steroids [42,43].

Despite the evident role of the ECS in neurodevelopment, there are few studies on the implication of the endocannabinoid system in perinatal programming, most of which have focused on a hypercaloric diet and the alteration of cannabinoid receptors in the short and long term in offspring, specifically in the brain and peripheral tissues [13,44,45,46,47,48].

Regarding maternal caloric restriction, Ramirez-Lopez and collaborators demonstrated that moderate caloric restriction during pregnancy alters endocannabinoids and/or endocannabinoid-related lipids in the hypothalamus of newborn pups. Specifically, the male offspring of mothers with caloric restriction showed reduced levels of arachidonic acid (AA) and oleoylethanolamide (OEA) without changes in the levels of the endocannabinoids 2-AG and anandamide (AEA), while females showed reduced levels of 2-AG and palmitoylethanolamide (PEA) when the restriction was implemented only during pregnancy. However, endocannabinoid levels decrease in males when caloric intake is reduced during pregestation [12,19]. Consistently, our results showed that the same maternal caloric restriction protocol implemented since pregestation induces effects on the gene expression of the synthesis and degradation enzymes of the main endocannabinoids (*Dagla, Daglb, Magl* and *Nape-pld*) in the hypothalamus of pups at PND2. The increase in the expression of *Dagla* and *Daglb* in females born to mothers with low caloric intake during pregestation/gestation could be due to a compensatory mechanism resulting from the decrease in 2-AG levels in the hypothalamus of these females, as seen in the study by Ramirez-Lopez et al. As stated above, if a persistent overactivity of these enzymes is prolonged during postnatal development in the absence of food restriction, they might eventually induce overeating and adiposity. Further research is needed to confirm this finding, particularly in the astrocyte compartment.

The main endocannabinoids anandamide (AEA) and 2 arachidonoylglycerol (2-AG) are synthesized from phospholipids containing arachidonic acid (AA). Fatty acids are fundamental nutrients during intrauterine development, with long-chain PUFAs (CL) such as arachidonic acid (AA) being one critical for fetal growth and neurodevelopment [37] The oncentration of AA depends on diet and endogenous synthesis, so malnutrition during pregnancy could alter the proper functionality of the enzymes involved in LM [49,50]. Interestingly, here we show that a maternal caloric restriction of 20% produces changes in the gene expression, but not the protein levels, of the main enzymes involved in LM (Scd1c, Acox1, Fasn, Srebf1 and Srebf2) in the hypothalamus of offspring at PND2. Specifically, we found an increase in the levels of monounsaturated fatty acid synthesis (Scd1) and of the first enzyme in fatty acid beta-oxidation (acox1), as well as the regulators of fatty acid metabolism Srebf1 and Srebf2 in the hypothalamus of offspring born to mothers with caloric restriction during pregestation/gestation. This dysregulation of the main enzymes involved in LM, as well as their regulators, could be due to the deficiency of fatty acids ingested by caloric-restricted mothers, and could explain the changes in the levels of endocannabinoids and their precursor AA mentioned above.

Malnutrition (overnutrition or undernutrition) early in life can have an impact on astrocytes. In general, both overnutrition and undernutrition increase GFAP expression in offspring [28]. In accordance with this, our results indicate that maternal caloric restriction induces astrogliosis (increases in Gfap and vimentin) in the hypothalamus of offspring.

Astrocytes undergo morphological and metabolic changes in response to maternal nutrition, and most studies have focused on structural changes with a focus on GFAP expression, associating it with various neuropathologies [28,51,52]; however, functional alterations in astrocytes can take place without changes in GFAP. As such, the functional consequences of maternal diet-induced alterations in astrocytes remain unclear.

Given the role of astrocytes in the control of neuroinflammation and metabolism, partly due to cannabinoid signaling, we hypothesized that the endocannabinoid system and LM of astrocytes play a key role in perinatal programming [21,23,53]. To further examine the role of astrocytes in the control of LM and cannabinoid signaling in the hypothalamus, and determine their degree of involvement in perinatal programming, we performed primary cultures of hypothalamic astrocytes of PND2 pups from control and dam undergoing moderate caloric restriction during pregestation and gestation.

Lipids are crucial in neurodevelopment, but neurons are ineffective in lipid synthesis. However, astrocytes are very active in de novo lipid synthesis and lipid transport to neurons. For example, astrocytes produce cholesterol and elongate and desaturate fatty acid precursors to synthesize polyunsaturated fatty acids (PUFAs, like DHA and AA). Furthermore, astrocyte lipid uptake through lipoprotein lipase is key to controlling energy homeostasis [23,28]. Similarly, a reduction in maternal caloric intake also induced effects on lipid metabolism enzymes (Acox1 and Fasn) expressed in astrocytes, excepting for Scd1 and Srebf, which are not affected by astrocytes, similar to that seen in the total hypothalamus. Therefore, knowing that astrocytes metabolize lipids to send metabolic signals to neurons [23], we propose that the main alterations in hypothalamic LM induced by prenatal caloric restriction are largely due to the changes in astrocytes affected by maternal malnutrition.

Regarding cannabinoid signaling in hypothalamic astrocytes, we found an effect of maternal caloric restriction on the genetic expression of Dagla, Magl, Nape-pld and Faah, coinciding with the results described for the total hypothalamus, except for Faah. In addition, the changes in both cases were more evident in females, indicating a marked sexual dimorphism.

Curiously, despite the fact that an increase in CB1 and CB2 receptors has been demonstrated in the hypothalamus of adult male rats born to mothers with moderate caloric restriction during pregestation and gestation [14], our results indicate that the cannabinoid receptors, CB1 and CB2, were unchanged in their gene and protein expression in the total hypothalamus and hypothalamic astrocytes of PND2 offspring. However, the increased expression of the endocannabinoid synthesis and degradation enzymes in astrocytes could indicate an increase in cannabinoid tone, which does not coincide with that described for the total hypothalamus. Therefore, it would be interesting to measure endocannabinoid production in astrocytes in future studies.

In addition, cannabinoid receptors are present in astroglial mitochondria, where they control respiration and cell energy processes [54]. A very interesting hypothesis for the cell substrate of maternal programming might be a differential expression of mitochondrial CB1 receptors induced by caloric restriction during development. This hypothesis will be considered in further studies.

Our results for cultured astrocytes were not totally consistent with those for the total hypothalamus. There is an important limitation of the in vitro studies derived of the abnormal environment on which cells are cultured. It is true that extracellular matrix proteins, types of cell-cell interaction (i.e., astrocyte-neuron contacts), influences of plasma hormonal/nutrients oscillations and other structural factors make it difficult to directly compare in vivo and in vitro approaches. We recognize this limitation and propose alternative approaches such as astrocyte-specific immunolabeling (GFAP) co-expressing relevant ECS components and lipid metabolism enzymes, as well as using transgenic mice lacking endocannabinoid 2-AG and AEA-hydrolysis enzymes Magl and Faah, in order to provide useful information about the specific role hypothalamic astrocytes play in caloric restriction during development.

In conclusion, the present study demonstrates that a moderate maternal caloric restriction implemented from pregestation and throughout gestation can induce changes in lipid/cannabinoid signaling of offspring hypothalami at PND2. Furthermore, we have demonstrated for the first time that these changes are mainly due to the metabolic control of hypothalamic astrocytes. Changes in cannabinoid signaling in hypothalamic astrocytes with an increase in the degradation enzymes Magl and Faah in females from caloric-restricted mothers could be related to the decrease in cannabinoid tone in the hypothalamus seen in the studies of Ramírez-López. Furthermore, these cannabinoid alterations, together with the increases in the oxidation signs of fatty acids (and an increase in Cpt1a in the same females) could be associated with weight loss and possibly with lower intake/appetite of offspring.

Therefore, our results point to astrocytes as fundamental pieces in nutritional programming, at least in the short-term metabolic alterations induced by perinatal nutritional injuries. Thus, the precise regulation of astrocyte signaling pathways emerge as promising targets for neuroregulation and/or neuroprotection from brain development.

## 4. Materials and Methods

### 4.1. Ethics Statement

All procedures were conducted in strict adherence to the principles of laboratory animal care (National Research Council, Neuroscience CoGftUoAi, Research B, 2003) following the European Community Council Directive (86/609/EEC) and were approved by the Ethical Committee of the University of Málaga. Special care was taken to minimize the suffering and number of animals needed to perform the procedures.

### 4.2. Animal Model and Diets

Adolescent female Wistar rats (Harlan, Barcelona, Spain) were individually housed in standard cages and maintained in controlled room conditions: 21 ± 1 °C room temperature, 40 ± 5% relative humidity and a 12-h light-dark cycle (lights off at 8:00 p.m). Prepubertal females were handled and allowed to acclimate for at least 3 weeks before diet assignation (C, control diet or R, caloric restriction diet). Two weeks before mating (pregestational period), animals were weighed and randomly assigned to a control (*n* = 6) or caloric restriction diet (*n* = 6; Figure 7). At this stage, no statistically significant difference in body weight among groups was found.

Control dams (C) were given free access to standard chow (SAFE A04, Panlab, Barcelona, Spain) consisting of 19.3% protein, 72.4% carbohydrate, 8.4% lipids and a total energy content of 3.3 kcal g^−1^. Calorie-restricted dams (R) were given a daily amount of food corresponding to 80% of the caloric intake of control dams, which was adjusted according to the body weight (20% of caloric restriction). It has been demonstrated that this moderate food restriction schedule is sufficient to induce long-lasting alterations in offspring without affecting fertility [7,55]. Water was provided ad libitum in both animal groups. Food intake and weight were measured daily.

The pregestation period included 2 weeks on the assigned diet (C or R diet), after which females were allowed to mate with males of the same strain in their home cage for 24 h at the beginning of proestrus. The presence of a spermatozoa plug in the vaginal smear confirmed successful mating, and this was designated as gestational day 0 (GD0). During the gestation, rat dams were maintained on the same dietary paradigm as in the pregestational period. Weights were measured daily and female rats were maintained on the calorie-restricted diet until GD 20, 2 days prior to birth (GD 20), when the restricted diet ended. At postnatal day 2–3, offspring were sacrificed (Figure 7).

### 4.3. Sample Collection

At postnatal day 2–3, offspring were sacrificed by decapitation. Brains were collected, quickly frozen and stored at −80 °C (control male: *n* = 11; restriction male: *n* = 9; control female: *n* = 10; restriction female: *n* = 6. To reduce the litter effect, 1 rat per litter was randomly selected; the remaining rats were used for other studies). The hypothalamus, including the median eminence, arcuate nucleus and ventromedial nucleus, was precisely dissected from the base of the brain according to the rat brain atlas of Paxinos and Watson [56]. Hypothalami samples were stored at −80 °C until being used for mRNA and protein analysis.

### 4.4. Primary Cultures of Astrocytes

Primary astrocytes were derived from male and female Wistar rat pups at PND2 separately (control male: *n* =10; restriction male: *n* = 14; control female: *n* = 9; restriction female: *n* = 10), as described previously [57]. Briefly, primary cultures were generated from the hypothalamus and maintained in DMEM:F12 (Gibco) with 10% fetal bovine serum (FBS) and 1% antibiotic antimycotic solution. After 10 days in vitro, when astrocyte cultures were 70–80% confluent, other cell types were removed by orbital rotation for at least 16 h at 280 rpm at 37 °C. This procedure results in enrichment of astrocytes to a purity of at least 95%. The remaining cells were plated at a density of 4.35 × 105 cells/cm^2^ and grown for 24 h. They were then grown in serum free media for 24 h before freezing.

### 4.5. RNA Isolation and Real-Time Quantitative PCR Analysis

Real-time PCR was performed by using specific sets of primer probes from TaqMan Gene Expression Assays (ThermoFisher Scientific, Waltham, MA, USA; Table 1). Hypothalami and astrocytes were homogenized on ice and RNA was extracted following the Trizol method according to the manufacturer’s instructions (ThermoFisher Scientific). A melting curve analysis was performed to ensure that only a single product was amplified. After analyzing several control genes, values obtained from the samples were normalized in relation to Actb (hypothalami) or b2microglubulin (astrocytes) levels that were found not to vary significantly between groups.

### 4.6. Western Blot Analysis

The hypothalami and astrocyte cultures (*n* = 6) were homogenized in 500 mL of ice-cold lysis buffer containing Triton X-100, 1 M 4-(2-hydroxyethyl)-1-piperazineethanesulfonic acid (HEPES), 0.1 M ethylenediaminetetraacetic acid (EDTA), sodium pyrophosphate, sodium fluoride (NaF), sodium orthovanadate (NaOV) and protease inhibitors using a tissue-lyser system (Qiagen). After centrifuging at 26,000× *g* for 30 min at 4 °C, the supernatant was transferred to a new tube. The Bradford method was used to measure the protein concentration of the samples. A quantity of 30 μg of each total protein sample was separated on 4–12% polyacrylamide gradient gels. The gels were then transferred onto nitrocellulose membranes (Bio-Rad Laboratories, Hercules, CA, USA) and stained with Ponceau red. Membranes were blocked in TBS-T (50 mM Tris-HCl pH 7.6, 200 mM NaCl, and 0.1% Tween 20) with 2% albumin fraction V from BSA (Roche, Mannheim, Germany) for 1 h at room temperature. The membranes were incubated overnight at 4 °C with the primary antibodies to the proteins of interest (Table 2). Mouse γ-adaptin was used as the reference protein. After several washes in TBS-T containing 1% Tween 20, an HRP-conjugated anti-rabbit or anti-mouse IgG (H + L) secondary antibody (Promega, Madison, WI, USA) diluted 1:10,000 was added, followed by incubation for 1 h at room temperature. After extensive washing in TBS-T, the membranes were incubated for 1 min with the Western Blotting Luminol Reagent kit (Santa Cruz Biotechnology, Santa Cruz, CA, USA), and the specific protein bands were visualized and quantified by chemiluminescence using a ChemiDocTM MP Imaging System (Bio-Rad, Barcelona, Spain). The results are expressed as the target protein/adaptin ratios.

### 4.7. Statistical Analysis

All data are expressed as the mean ± SEM. Animal model data were analyzed by two-way ANOVA (sex and maternal diet). IBM SPSS Statistics 23 and GraphPad Prism 6 software programs were used. Subsequent multiple comparisons between groups were carried out using Tukey adjustments or simple effect analysis (Fisher’s test) in cases of factor effect but no interaction. *p* < 0.05 was considered statistically significant.

## Figures and Tables

**Figure 1 ijms-22-06292-f001:**
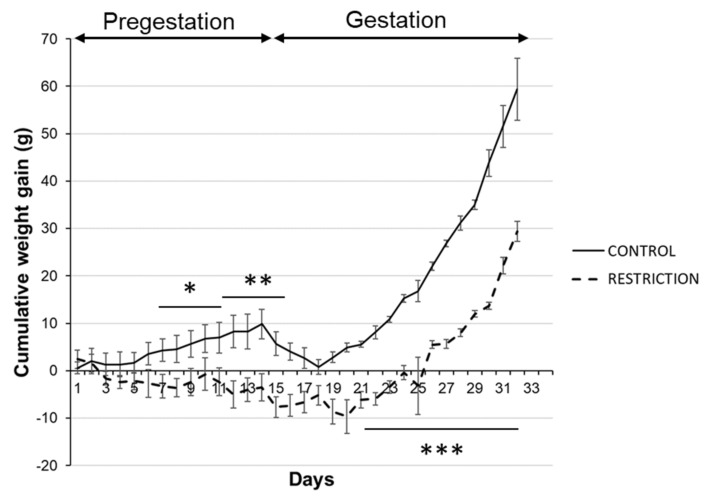
Maternal weight gain during pregestation and gestation periods. Cumulative weight gain (g) of control (continuous line) and restricted dams (dashed line) during pregestation and gestation. Values are expressed as means ± SEM. * *p* < 0.05, ** *p* < 0.01, *** *p* < 0.001.

**Figure 2 ijms-22-06292-f002:**
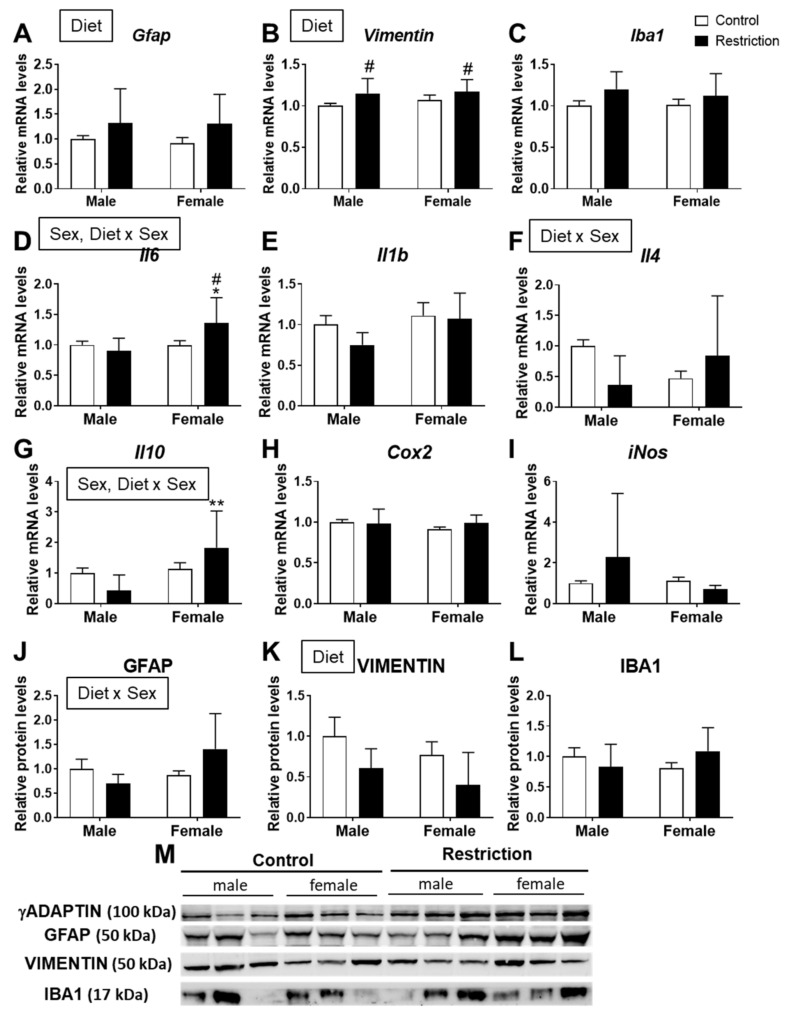
Effect of maternal caloric restriction on the relative mRNA (*Gfap* (**A**), *Vimentin* (**B**), *Iba1* (**C**), *Il6* (**D**), *Il1b* (**E**), *Il4* (**F**), *Il10* (**G**), *Cox2* (**H**) and *iNos* (**I**)) and protein (GFAP (**J**), VIMENTIN (**K**) and IBA1 (**L**)) levels of neuroinflammation-related genes in the hypothalami of male and female offspring at postnatal day 2. Representative membranes of the main changes (**M**). Data are expressed as the mean ± S.E.M. (control male: *n* = 11; restriction male: *n* = 9; control female: *n* = 10; restriction female: *n* = 6). Tukey or single-effect analysis: */** *p* < 0.05/0.01 vs. males (sex difference); # *p* < 0.05 vs. same-sex control group.

**Figure 3 ijms-22-06292-f003:**
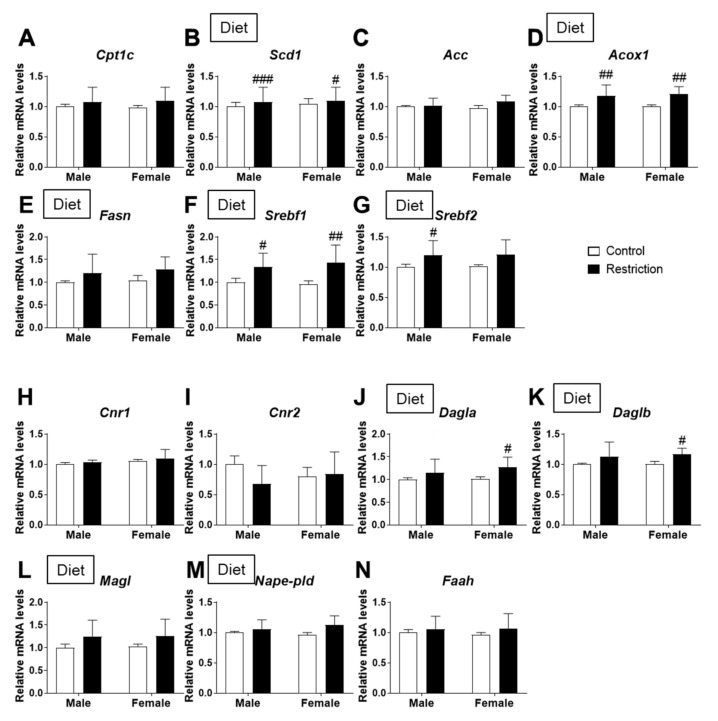
Effect of maternal caloric restriction on mRNA expression of principal genes of lipid metabolism: *Cpt1a* (**A**), *Scd1* (**B**), *Acc* (**C**), *Acox1* (**D**), *Fasn* (**E**), *Srebf1* (**F**) and *Srebf2* (**G**); and the endocannabinoid system: *Cnr1* (**H**), *Cnr2* (**I**), *Dagla* (**J**), *Daglb* (**K**), *Magl* (**L**), *Nape-pld* (**M**) and *Faah* (**N**) in the hypothalami of male and female offspring at postnatal day 2. Data are expressed as the mean ± S.E.M. (control male: *n* = 11; restriction male: *n* = 9; control female: *n* = 10; restriction female: *n* = 6). Tukey or single-effect analysis: #/##/### *p* < 0.05/0.01/0.001 vs. same-sex control group. The main results of the two-way ANOVA (effect of sex, diet and/or sex × diet interaction) are indicated above the graphs.

**Figure 4 ijms-22-06292-f004:**
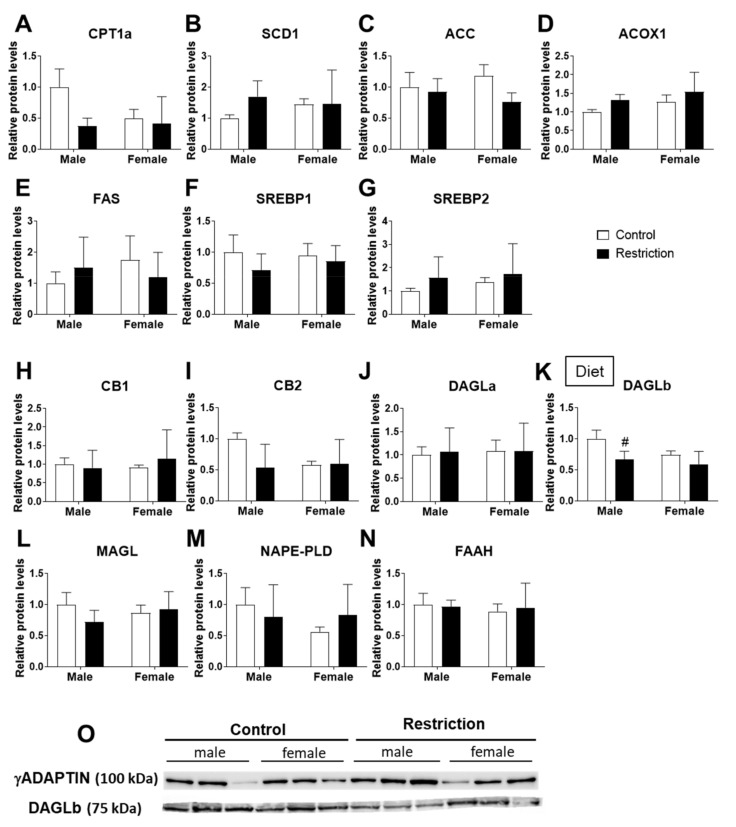
Effect of maternal caloric restriction on protein levels of principal factors of lipid metabolism: *Cpt1a* (**A**), *Scd1* (**B**), *Acc* (**C**), *Acox1* (**D**), *Fasn* (**E**), *Srebf1* (**F**) and *Srebf2* (**G**); and the endocannabinoid system: *Cnr1* (**H**), *Cnr2* (**I**), *Dagla* (**J**), *Daglb* (**K**), *Magl* (**L**), *Nape-pld* (**M**) and *Faah* (**N**) in the hypothalami of male and female offspring at postnatal day 2. Representative membranes of the main changes (**O**). Data are expressed as the mean ± S.E.M. (control male: *n* = 11; restriction male: *n* = 9; control female: *n* = 10; restriction female: *n* = 6. Tukey or single-effect analysis: # *p* < 0.05/0.01/0.001 vs. same-sex control group. The main results of the two-way ANOVA (effect of sex, diet and/or sex × diet interaction) are indicated above the graphs.

**Figure 5 ijms-22-06292-f005:**
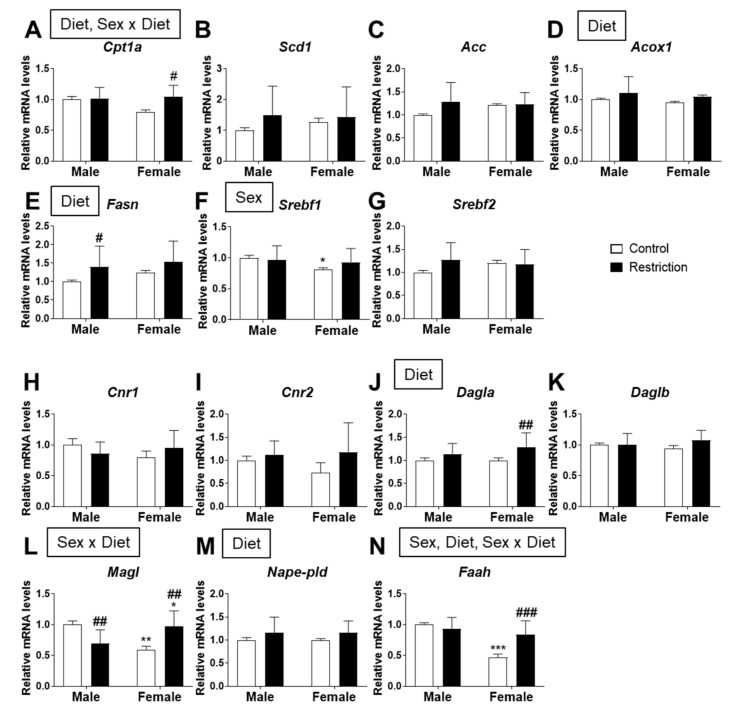
Effect of maternal caloric restriction on mRNA expression of principal genes of lipid metabolism: *Cpt1a* (**A**), *Scd1* (**B**), *Acc* (**C**), *Acox1* (**D**), *Fasn* (**E**), *Srebf1* (**F**) and *Srebf2* (**G**); and the endocannabinoid system: *Cnr1* (**H**), *Cnr2* (**I**), *Dagla* (**J**), *Daglb* (**K**), *Magl* (**L**), *Nape-pld* (**M**) and *Faah* (**N**) in hypothalamic astrocytes of male and female pups at PND2. Data are expressed as the mean ± S.E.M. (control male: *n* = 10; restriction male: *n* = 14; control female: *n* = 9; restriction female: *n* = 10). Tukey or single-effect analysis: */**/*** *p* < 0.05/0.01/0.001 vs. males (sex difference); #/##/### *p* < 0.05/0.01/0.001 vs. same-sex control group. The main results of the two-way ANOVA (effect of sex, diet and/or sex x diet interaction) are indicated above the graphs. We also analyze the effect of sex and maternal caloric restriction on the protein levels of the main enzymes of lipid metabolism and the endocannabinoid system. Two-way ANOVA showed an effect of sex on CPT1a [F (1,20) = 7.079; *p* < 0.05] and SREBP2 [F (1,20) = 5.877; *p* < 0.05] protein levels, with an increase in CPT1a in hypothalamic astrocytes from females born to caloric-restricted mothers compared to males from the same group (* *p* < 0.05; Figure 6A).

**Figure 6 ijms-22-06292-f006:**
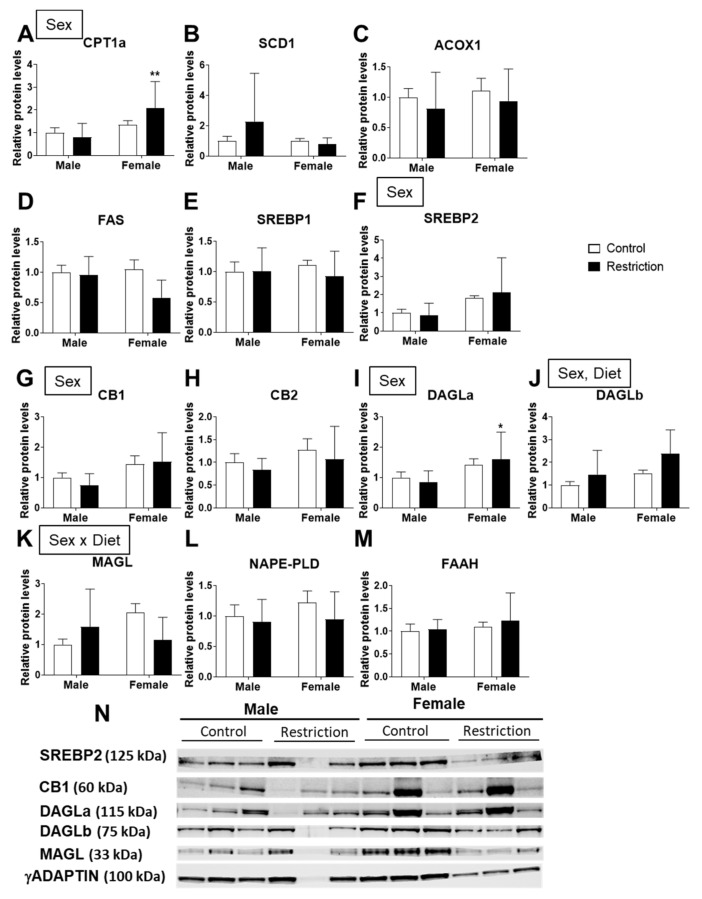
Effect of maternal caloric restriction on protein levels of principal genes of lipid metabolism: CPT1a (**A**), SCD1 (**B**), ACOX1 (**C**), FAS (**D**), SREBP1 (**E**) and SREBP2 (**F**); and the endocannabinoid system: CB1 (**G**), CB2 (**H**), DAGLa (**I**), DAGLb (**J**), MAGL (**K**), NAPE-PLD (**L**) and FAAH (**M**) in hypothalamic astrocytes of male and female pups at PND2. Representative membranes of the main changes (**N**). Data are expressed as the mean ± S.E.M. (control male: *n* = 10; restriction male: *n* = 14; control female: *n* = 9; restriction female: *n* = 10). Tukey or single-effect analysis: */** *p* < 0.05/0.01 vs. males (sex difference). The main results of the two-way ANOVA (effect of sex, diet and/or sex × diet interaction) are indicated above the graphs.

**Figure 7 ijms-22-06292-f007:**
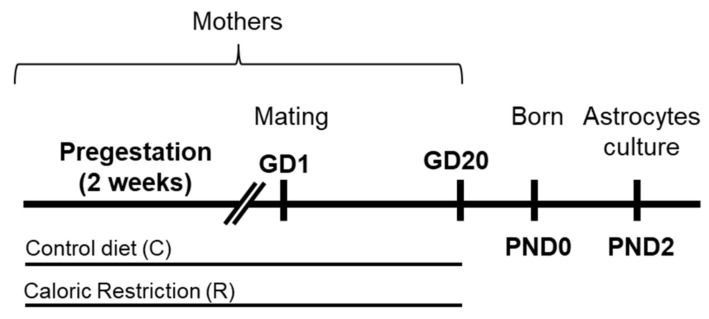
Experimental design. Female rats (dams) were exposed to a control diet (C) or caloric restriction diet (R) during pregestation (2 weeks before mating) and gestation (20 days approximately).

**Table 1 ijms-22-06292-t001:** Primer references for Taqman Gene Expression Assays.

Gene Symbol	Assay ID	GenBank Accession Number	Amplicon Length (bp)
*B2M*	Rn00560865_m1	NM_012512.2	58
*Actb*	Rn00667869_m1	NM_031144.3	91
*Srebf1*	Rn01495769_m1	NM_001276707.1	79
*Srebf2*	Rn01502638_m1	NM_001033694.1	61
*Cpt1c*	Rn01475546_m1	NM_001034925.2	88
*Cpt1a*	Rn00580702_m1	NM_031559.2	64
*Insr*	Rn00573576_m1	NM_022212.2	65
*Acox1*	Rn01460628_m1	NM_017340.2	63
*Acaca*	Rn00573474_m1	NM_022193.1	60
*Scd1*	Rn00594894_g1	NM_139192.2	86
*LepR*	Rn01433205_m1	NM_012596.1	94
*Fasn*	Rn01463550_m1	NM_017332.1	148
*Cnr1*	Rn02758689_s1	NM_012784.4	92
*Cnr2*	Rn01637601_m1	NM_020543.4	68
*Dagla*	Rn01454303_m1	NM_001005886.1	61
*Daglb*	Rn01453770_m1	NM_001107120.1	57
*Mgll*	Rn00593297_m1	NM_138502.2	78
*Napepld*	Rn01786262_m1	NM_199381.1	71
*Faah*	Rn00577086_m1	NM_024132.3	63
*Vimentin*	Rn00667825_m1	NM_031140.1	83
*Gfap*	Rn01253033_m1	NM_017009.2	75
*Aif1*	Rn03993468_g1	NM_017196.3	82

**Table 2 ijms-22-06292-t002:** Antibodies used for protein expression by Western blotting.

Antigen	Manufacturing	Dilution
ACC	Cell signal (C83B10) Rabbit mAb	1/1000
ACOX1	Abcam [EPR19038] ab184032	1/500
γ-adaptina	BD Biosciences. Mouse Adaptin 610385	1/2000
CPT1a	Abccam (ab102679) rabbit	1/1000
FAS	Cell Signaling (C18C12) Rabbit mAb	1/1000
SCD1	Abcam [CD.E10] ab19862	1/250
Daglα	Biorbyt #orb-158553	1/100
Daglβ	Biorbyt #orb-182975	1/100
CB1	Abcam #Ab23703	1/200
CB2	Abcam #Ab3561	1/200
NAPEPLD	Abcam #Ab95397	1/1000
FAAH	CAYMAN #101600	1/100
GFAP	Thermofisher #MA5-12023	1/100
VIMENTINE	Thermofisher #MA5-11883	1/100
SBREP1	Santa Cruz #sc-365513	1/100
SBREP2	Santa Cruz #sc-13552	1/100
IBA1	Wako	1/500
MGLL	Abcam #Ab24701	1/200

## Data Availability

The data that support the findings of this study are available on reasonable request from the corresponding author.

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
