# Peer review of "Analysis of Both Lipid Metabolism and Endocannabinoid Signaling Reveals a New Role for Hypothalamic Astrocytes in Maternal Caloric Restriction-Induced Perinatal Programming"

_ijms, 2021, doi:10.3390/ijms22126292_

Round 1
Reviewer 1 Report
In the present paper authors investigated the effect of maternal caloric restriction on lipid metabolism and endocannabinoid system function in hypothalamic astrocytes. Offsprings born to undernourished mothers presented with malformations in hypothalamic functions - enzymes involved in lipid metabolism and endocannabinoids' synthesis and degradation.
Figure legends should be rewritten in more precise manner.
Figure 2: figure is not showing just neuroinflammation-related genes. Define more precisely the model system.
Figure 3, 5 and 6. is there any gender-related difference in the amount of enzyme expression in the offsprings of undernourished mothers
- use of abbreviations in the abstract section
Reviewer 2 Report
This submission demonstrated a high degree of involvement of astrocytes in nutritional programming through rat model received moderate caloric restriction. Please conduct the concerns below.
- Diet can modify the endocannabinoid system (ECS) in brain during development is the main background of this hypothesis. Additionally, astrocytes are affected by early life adversity (ALS) in rodents. However, less messages of hypothalamic astrocytes in perinatal programming were introduced. Why?
- Sample size in figure legends shown N = 6-11 remained obscure. Please indicate each value per group in clear.
- The alterations induced by maternal caloric restriction in lipid metabolism and cannabinoid signaling from total hypothalamus of the offspring needs reference(s) to support.
- Similarly, abnormal hypothalamic development during offspring also needs the reference(s).
- Association of ECS and development was not discussed, particularly the sex difference.
- Findings in cultured astrocytes were not consistent with that in the total hypothalamus. Please speculate the way to solve this problem in discussion.
- In conclusion, changes were mentioned mainly due to the metabolic control of hypothalamic astrocytes. However, reasons(s) and mechanism(s) were not conducted. Why?
- Limitation(s) of current report were ignored.
Round 2
Reviewer 2 Report
This submission has been improved according to comments.